# Prospective Evaluation of Transsphenoidal Pituitary Surgery in Patients with Cushing’s Disease: Delayed Remission and the Role of Postsurgical Cortisol as a Predictive Factor

**DOI:** 10.3390/healthcare12181900

**Published:** 2024-09-22

**Authors:** Athanasios Saratziotis, Maria Baldovin, Claudia Zanotti, Sara Munari, Diego Cazzador, Enrico Alexandre, Luca Denaro, Jiannis Hajiioannou, Enzo Emanuelli

**Affiliations:** 1Department of Otolaryngology, University Hospital of Larissa, 41334 Larisa, Greece; jhajiioannou@gmail.com; 2Otorhinolaryngology Unit, San Martino Hospital, ULSS1 Dolomiti, 32100 Belluno, Italy; maria.baldovin@gmail.com; 3Otolaryngology Unit, S. Valentino Hospital, Montebelluna (TV), AULSS2 Marca Trevigiana, 31100 Treviso, Italy; zanotti.claudia89@gmail.com (C.Z.); saramunari@hotmail.it (S.M.); 4Department of Neurosciences, Otolaryngology Section, University of Padua, 35122 Padova, Italy; gkmcaz@hotmail.it; 5Ear Nose Throat (ENT) Audiology and Phoniatric Unit, Department of Surgical Pathology, Medical, Molecular and Critical Area, Azienda Ospedaliero-Universitaria Pisana, University of Pisa, 56124 Pisa, Italy; alexandre.enrico@gmail.com; 6Department of Neurosciences, Neurosurgery Section, University of Padua, 35122 Padova, Italy; luca.denaro@unipd.it; 7Otolaringology Unit, Ca’ Foncello Hospital, Local Health Unit N.2 “Marca Trevigiana”, 31100 Treviso, Italy; enzoemanuelli@libero.it

**Keywords:** Cushing’s disease, ACTH-secreting pituitary adenoma, endoscopic transsphenoidal surgery, treatment outcome, postoperative cortisol, remission outcome

## Abstract

Background. Transsphenoidal surgery is the treatment of choice for Cushing’s disease. Successful surgery is associated with subnormal postoperative serum cortisol concentrations and cortisoluria levels, which may guide decisions regarding immediate reoperation. Remission is defined as the biochemical reversal of hypercortisolism with the re-emergence of diurnal circadian rhythm. Methods. A single-center prospective cohort study was conducted among thirty-three patients who underwent transsphenoidal pituitary surgery for Cushing’s disease. Postoperative surgical outcomes, daily morning cortisolemia, and 24 h urinary-free cortisol from the first to the fifth morning were evaluated. Results. All patients underwent surgery, with a remission rate of 81.2%. Of the 26 patients who achieved early remission, 92% remained in remission. Two patients (7.7%) showed recurrence of Cushing’s disease during a mean follow-up of 81.7 months. Early postoperative hypocortisolism suggests complete removal of the tumor, correlating with high rates of remission (*p* < 0.001). Also, in 12.5% of patients with early cortisol values >138 nmol/L, there was a gradual late remission. Conclusions. In our cohort of patients, the endoscopic transsphenoidal approach was safe and effective in the treatment of Cushing’s disease. We demonstrated that serum and urinary cortisol concentrations did not experience significant fluctuations from the first to the fifth day. This constitutes an accurate predictor of durable remission, comprising a distinctive finding in the intermediate term by our team.

## 1. Introduction

Adrenocorticotropic hormone (ACTH)-secreting pituitary tumors cause hypercortisolism, also known as Cushing’s disease (CD). Significant morbidity and mortality are associated with this endocrinopathy. An ACTH-secreting pituitary microadenoma is the most common cause of CD (70%). Transsphenoidal surgery (TSS) is highlighted as an effective first-line therapy for Cushing’s disease, with remission rates ranging from 70% to 90% at experienced pituitary centers [1,2,3,4,5,6,7,8].

Following successful corticotropinoma resection, the remaining normal pituitary corticotrophs are suppressed by long-standing hypercortisolaemia, leading to transient postoperative adrenocorticotropin (ACTH) and cortisol deficiency [9,10]. Cortisol and adrenocorticotropin administered intravenously have a short half-life in serum, leading to the expectation of a rapid drop in their levels during the immediate postoperative period [11,12]. Total hypophysectomy achieves a faster rate of reduction in cortisol serum levels than partial hypophysectomy, as has been previously demonstrated [13]. 

It is thought that the presence of residual tumor cells reflects a failure to achieve hypocortisolemia, and cortisol serum is considered as a biochemical marker of early surgical remission [14,15]. Although experienced centers report high success rates, early remission does not eliminate a recurrence rate of 2–35%, even in the long run. Recurrence has been observed up to thirty years postoperatively [14,15,16,17].

Characteristics of the adenoma (aggressivity and spread, histology, size, identification of the adenoma through magnetic resonance imaging, etc.), surgical techniques, experience of the surgeon, and biochemical criteria represent CD remission predictors that have been taken into consideration in the literature [18].

Cortisol concentration in the immediate postoperative period has been the most widely used long-term prognostic indicator in most of the published series [19,20,21,22,23]. CD remission is defined by morning cortisol <5 μg/dL (<138 nmol/L) or urinary-free cortisol (UFC) <10–20 μg/24 h (<28–56 nmol/24 h) at 7 days after the operation, based on the guidelines of the management of CD published in 2015 [14]. According to some authors, the optimal times to determine cortisol levels are 24–48 h, 7–14 days, or even months after surgery [19,20,21,22,23,24]. Plasma ACTH is useful after surgery as a prognostic indicator in CD, but only a few series have analyzed this issue. After the complete removal of the pituitary adenoma in patients with CD, considering the fact that ACTH half-life is about 10 min, a reasonable assumption would be that a marked decrease in plasma ACTH values would be detected in the first 12–24 h [25,26,27]. 

This study aims to determine the outcome of the operation, along with the performance of the postoperative evaluation and monitoring of patients with any type of complications (surgical and/or endocrinological). In addition to tumor size, age at surgery and months of follow-up, preoperative levels of ACTH and morning cortisolemia and cortisoluria were also assessed to determine if there were statistically significant differences between males and females, and between macro- and microadenomas, and also to evaluate if any of these factors could be associated with the outcome of the disease after surgery.

## 2. Materials and Methods

### 2.1. Selection of Patients

This is a prospective cohort study conducted at the University Hospital of Padova, Department of Neuroscience, Otolaryngology Section, from June 2012 to July 2023. This is a referral center for 3.5 million inhabitants and the only center in the region for pituitary surgery. 

We studied patients diagnosed with Cushing’s disease involving ACTH-secreting pituitary adenomas. The study protocol was authorized by the Ethics Committee of the University Hospital of Padova and given the approval of the institution’s research with protocol number 32358/03-2012. Patient data were extracted from electronic medical records and an MRI picture-archiving and communication system.

Pediatric patients, adult patients unfit for surgery due to comorbidity or lack of follow-up and patients not completely tested according to the protocol were excluded. The study involved all those patients who agreed to be monitored exclusively by our center throughout the postoperative period and not patients who would be incapacitated in the long run for any reason. Seven patients had previously undergone surgery in another center. All patients were placed on a protocol according to the guidelines for Cushing’s disease and none were taking drugs that could alter the intra- and post-intervention measurements. For all seven patients, one year had passed since the first operation.

Eligible patients were included after obtaining their informed consent. A trans-naso-septal transsphenoidal approach was used, and all operations were performed by the same surgical team, including otolaryngology and neurosurgeon specialists. The following variables were evaluated: age at surgery, months of follow-up, morning basal cortisol levels, preoperative ACTH and 24 h urinary-free cortisol, from the first to the fifth day postoperatively. These variables were studied by comparing patient subgroups (remission/non-remission, male/female). 

### 2.2. Preoperative Diagnosis of Cushing’s Disease 

In our study, Cushing’s disease was confirmed on the basis of elevations of at least two of the following laboratory tests: late-night salivary cortisol, 24 h urinary-free cortisol or low-dose dexamethasone suppression testing (1 mg overnight or 2 mg over 48 h, normal cortisol < 1.8 μg/L). Supplemental diagnostic testing included midnight serum cortisol and morning cortisol following 8 mg dexamethasone suppression testing (HDDST) overnight.

According to guidelines, in patients with ACTH-dependent hypercortisolism, at least one of the following supplementary exams is necessary: the response of serum cortisol or ACTH to ovine corticotropin-releasing hormone (oCRH) stimulation testing, adenoma identification by magnetic resonance imaging of the pituitary (MRI), or inferior petrosal sinus sampling exam (IPSS) [28,29,30,31]. 

In our study, the adenoma was not clearly visible under MRI in two patients, so the catheterization of the lower petrosal sinuses was performed (IPSS) and the disease was confirmed.

Based on size, the adenomas were divided into microadenomas (up to 9 mm), macroadenomas (between 10 and 39 mm), and giant adenomas (>40 mm). Demographic data of the patients are presented in Table 1.

### 2.3. Postoperative Management

Postoperatively, an early-control brain CT was performed (within 12 h) to rule out surgical complications. Patients were monitored through 24 h evaluation, and the samples for the measurement of early postoperative cortisolemia levels were collected at 7–9 a.m. In no case was corticosteroid therapy administered during surgery. 

To prevent adrenal crisis, patients began therapy on the third postoperative day with cortisone acetate, which is easier to continue at home, only after measuring cortisol levels. If cortisolemia values were <270 nmol/L, the treatment was continued at home. In the cases where substitution was given, cortisone acetate was used in doses of 12.5–37.5 mg daily. The decision to continue cortisone substitution was then made by the evaluating clinician, based on clinical signs and biochemical results.

Τhe small increases in cortisolemia and cortisoluria after the administration of cortisone acetate were considered corrected values without potential interference in our evaluation. If the values were between 270 and 500 nmol/L, therapy was considered necessary only in special cases, for example in patients undergoing reoperation or in the presence of stressful events or fever. Therapy was not required if levels were >500 nmol/L (18 μg/dL), indicative of disease persistence.

In our laboratory, urinary cortisol was measured by tandem mass spectrometry (LC-MS/MS), and serum cortisol by immunometric method. 

The function of the pituitary–adrenal axis was studied through measurements of morning cortisolemia and urinary cortisol. Data were collected within 8 days of surgery and after the last endocrinological follow-up. In the case of disease persistence after surgery, late hormonal data were not affected by the medical therapy applied for Cushing’s disease, which in no case had yet been started.

Serum cortisol levels were routinely obtained at 6 h intervals from the first postoperative day (POD-1) until the fifth day (POD-5).

On the first postoperative day, serum and urine cortisol was obtained at 8:00 h or 12:00 h, depending on the surgery completion time. For the next postoperative days measurements were made at 7–9 a.m. Daily morning cortisol levels were evaluated and compared from POD-1 to POD-5 during the in-hospital stay. If a nadir serum cortisol level of <5 μg/dL by POD-5 was detected, an early biochemical remission was assigned, before the need for the administration of exogenous glucocorticoids was established, as well as the necessity for glucocorticoid replacement because of sustained hypocortisol.

In accordance with the 2015 Guidelines, we established CD remission as morning cortisol <5 μg/dL (<138 nmol/L) or urinary-free cortisol (UFC) <10–20 μg/24 h (<28–56 nmol/24 h). 

### 2.4. Statistical Analysis

Data were analyzed using the SPSS program (SPSS Inc., SPSS Statistics for Windows, Version 17.0, Chicago, IL, USA). The distribution of the variables was studied using the Shapiro–Wilk normality test. 

Patient subgroups (remission/non-remission, male/female) were compared with the following variables: age at surgery, months of follow-up, cortisolemia, cortisoluria and preoperative ACTH, and postoperative morning cortisolemia and cortisoluria during the same five-day period. The comparison was performed by using the Mann–Whitney non-parametric test for the non-normal distribution of the variables. 

Statistical significance was assigned to values of *p* < 0.05. Correlations regarding all cortisol measurements between the first and the fifth day were analyzed using the Spearman test.

## 3. Results

From June 2012 to July 2023, a total of 39 patients were evaluated. Six patients (four children and two adults) affected by several comorbidities and unfit for surgery were excluded from the protocol. Finally, 33 patients, 26 females (78.8%) and 7 males (21.2%), aged 41.67 years on average (range 15–67 years), underwent the removal of an ACTH-secreting pituitary adenoma through the endoscopic endonasal approach. Demographic data are reported in Table 1.

Of the 33 patients, 26 had not undergone previous treatments. Seven patients were admitted to our center with recurrence or persistence of the pituitary adenoma after one or more treatments (surgery, radiotherapy, chemotherapy). The mean follow-up was 81.7 months (range 59–132 months). Complications that occurred after surgery were as follows: deep vein thrombosis (n = 2), rhinoliquoral fistula (n = 2), fever (n = 3), and Aspergillus meningoencephalitis (n = 1). In one patient, fever was associated with cerebrospinal fluid (CSF) leak. The patient was given intravenous antibiotic therapy and underwent surgical repair of the fistula. In the other two cases, it was not associated with specific infection or meningitis. The main endocrinological complications were transient diabetes insipidus (n = 12), hypothyroidism (n = 4), growth hormone (GH) deficiency (n = 3), anterior panhypopituitarism (n = 1), and anterior and posterior panhypopituitarism (n = 2). After surgery, remission was achieved in 81.2% of patients. One patient was lost in follow-up and was therefore excluded from the comparison of variables for disease remission. Of the 26 patients who achieved early control, 92% remained in remission. 

Six patients (18.7%) experienced disease persistence after surgery. None of these underwent early re-intervention, as some authors suggest, since in five cases the adenoma had been identified intra-operatively and the surgeons had removed all the visible tumor tissue; in one case, however, the tumor had spread to the cavernous sinus, and as much adenoma tissue as possible was removed safely. 

In two patients (7.7%), the disease relapsed after initial remission: at 48 months after surgery in one case, and at 60 months in the other. In both patients there were indications for surgical re-intervention. In the first case, a pituitary microadenoma was identified, with early postoperative cortisolemia of 120 nmol/L; in the second case, by contrast, a macroadenoma was diagnosed, with early postoperative cortisolemia of 320 nmol/L. Neither had undergone previous adenoma surgery.

Gradual remission was observed in four patients who were initially considered patients not in remission, with morning cortisol levels greater than >138 nmol/L. After the first month, morning cortisol levels dropped to <138 nmol/L, and they were considered patients with late remission. 

### 3.1. Laboratoy Analysis

Regarding the differences between males and females, preoperative ACTH was significantly higher in males than in females (*p* = 0.023). A difference, although not statistically significant, can also be seen by comparing the variable of tumor size, with higher values in the macroadenomas (*p* = 0.07). However, no statistically significant difference emerged with respect to the other variables analyzed between the subgroups. Biochemical data analyzed by gender are shown in Table 2.

When patients were divided into remission and non-remission groups, ACTH concentrations in the non-remission group (recurrence group) were also slightly lower than in the remission group (recurrence group—50.0 [39.0–100] (ng/mL); remission group—56 [46.0–101.0] (ng/mL), *p* value 0.59). Biochemical data analyzed by remission and non-remission of the patients are shown in Table 3.

### 3.2. Preoperative MRI Classification

Based on the preoperative MRI of the pituitary gland, 23 were classified as microadenomas (of which 2 were not visible on MRI), and 10 as macroadenomas. The median of the age (years) of the patients at surgery and the months of follow-up analyzed by tumor size are shown in Table 4.

### 3.3. Remission Group versus Recurrence Group

Postoperatively, cortisolemia was evaluated between 7 and 9 a.m. on the first five postoperative days, and in any case, at least 6 h after taking cortisone acetate, if prescribed.

In the remission group, the median of postoperative early cortisolemia values was 58.5 nmol/L (range 43.0–120.0 nmol/L), while in the group in which the disease persisted, the median was 453.5 nmol/L (range 304–704 nmol/L), with a significant difference between the two groups (*p* < 0.001) (Table 2). 

However, since this is a sample with limited numbers of participants (also due to the fact that the disease is rare), the values obtained cannot be considered as significant, and therefore cannot be used as valid cut-off values for the entire patient population.

According to the cut-off of early cortisolemia (138 nmol/L is the value most often used in the literature [25,26,27,28,29,30]), which defines the disease in remission, a high sensitivity of 100% and an acceptable specificity of 81% are noted in our case. 

Using the cut-off value of 138 nmol/L, 81.25% of patients achieved early remission after surgery. Analyzing hormonal levels, we found that hypocortisolemia in the early postoperative period was strongly correlated with surgical success (*p* < 0.001) in the long term.

As for 24 h cortisoluria (UFC), it was significantly lower (*p* = 0.019) in patients in remission, at 40.5 (28.5–98.0) nmol/24 h, than in patients with persistence of disease, at 185.5 (72.0–315.0) nmol/24 h. However, it did not vary significantly between males and females, or between micro- and macroadenomas.

In the evaluation of all the examined parameters (blood and urine) from the first to the fifth day, no significant fluctuations in cortisol levels in blood and urine were observed (shown in Figure 1 and Table 5). 

We also found that there was a positive correlation in all the above parameters between the first and second, third and fourth, and first and fifth days in the remission and non-remission groups. This means that as one value increased, the other value increased too when they were compared across all five days (Table 6).

### 3.4. Histopathological Results

Factors that have been studied as predictors of the outcome of the disease also include the intra-operative identification of adenoma and the confirmation of corticotropic adenoma on histological examination. In 8 of 32 cases, the histological examination of the collected material did not reveal the presence of pituitary adenoma; a microadenoma was identified in all cases, including the two adenoma cases not visible in preoperative imaging. All of the aforementioned patients have experienced disease remission following surgery and remain disease-free at the last follow-up. The characteristics of the four patients with late remission are described in Table 7.

However, in six cases, a microadenoma was diagnosed on preoperative MRI, while in two cases the adenoma was not visible on imaging but the diagnosis was confirmed by preoperative catheterization of the petrous sinuses. In all cases, the adenoma was seen during surgery. Disease remission without histological confirmation may be explained by postoperative necrosis, fragility of the surgical material that may deteriorate during transport to the pathology department, or aspiration of the tumor during surgery. 

### 3.5. Postoperative MRI Data 

The first MRI with postoperative contrast medium was performed four months after surgery, and then, in the absence of suspicion of disease recurrence, annually. Radiological remission was considered as radical (gross total resection) if there was no evidence of tumor residue at the follow-up MRI; it was considered as total, subtotal, partial and insufficient if residues of <5%, <20%, 20–50% and >50% were observed under MRI, respectively [30,31]. In our study, four patients were found to have residual tissue with subtotal remission (<20%), while there were four patients with doubtful residual pathological tissue (<5%).

## 4. Discussion

In our study, remission was achieved in 81.2% of patients in the first four years following surgery. Persistence of the disease after surgery was experienced by 18.7%. The disease relapsed after initial remission in two cases: at 48 months after surgery in one, and at 60 months after surgery in the other.

Of the six patients with disease persistence after surgery, one underwent a bilateral adrenalectomy due to the poor control of the disease with medical therapy (currently in remission, without Nelson’s syndrome); one was treated with conventional radiotherapy and subsequent medical therapy, and four are still being treated with medical therapy with good control of symptoms.

The transsphenoidal endoscopic approach (TSA) is the initial treatment of choice for ACTH-secreting pituitary adenomas. Τhe most significant complications we faced were deep vein thrombosis, rhinoliquoral fistula, meningoencephalitis, and (in one patient) transient diabetes insipidus. Other major complications of interest included the postoperative syndrome of inappropriate antidiuretic hormone secretion, permanent diabetes insipidus, CSF leakage, carotid artery injury, epistaxis, meningitis, and vision changes. In our study, the findings regarding the severity of surgical complications as well as their number in all operated patients are compatible with those from the literature on this technique [7,21,27,31,32,33]. In all our patients, a microadenomectomy was always performed as the first choice. Total hypophysectomy should be performed in selected cases, limited to patients who are no longer fertile and whose pituitary function is already severely compromised, meaning that the approaches available have already been tested without success [17].

Postoperative remission rates of 70–98% for TSA in CD have been described in numerous retrospective studies [16]. The success or failure of surgery for CD is attributed to a number of factors, such as tumor size, prior surgery, visibility of tumor on MRI, surgical technique, and cavernous sinus invasion [34].

The surgical experience of the team and the volume of cases of the center justify the variability of the remission rates reported in the literature (from 65% to 97%). Recently, the development of a commonly accepted definition of early biochemical remission has become necessary to try to standardize the effects of treatment. In addition, in a subgroup of patients who achieved early postoperative biochemical remission, there was a subsequent recurrence of hypercortisolemia in a long-term follow-up [5,16,34,35].

Several factors have been studied as predictors of remission, including tumor size (microadenoma versus macroadenoma), location, invasiveness, tumor visualization in preoperative imaging, intra-operative identification of the adenoma, and the confirmation of corticotropic adenoma on histological examination; however, there is no unanimous agreement in the literature [5,7,8,18,32,34,36].

ACTH-secreting pituitary tumors are known to be more frequent in females, although the disease has more aggressive characteristics in males, in terms of both hormone levels, and prevalence and seriousness of complications [37]. This aspect is also confirmed in our study, in which, by analyzing the different variables between males and females, it emerged that the preoperative ACTH was significantly higher in males than in females, suggesting that although the disease is more frequent in females, males are more hormonally aggressive. This fact suggests a more pronounced secretory activity of the pituitary adenoma in males. After the removal of a pituitary adenoma, hormone levels often stabilize and return to normal levels, regardless of gender. This happens because the goal of the surgery is to remove the adenoma that disrupts the normal function of the pituitary gland, thereby allowing hormone levels to return to normal. 

Microadenomas are found in the majority of patients with CD; however, they can be localized and removed effectively. A difference, although not statistically significant, can also be seen by comparing the variable of tumor size, with higher values in the macroadenomas. In our study, 23 microadenomas and 10 macroadenomas were identified. Here, 18 of 22 (81%) microadenomas and 8 of 10 (80%) macroadenomas went into remission after surgery. One patient (1 of 23) with a microadenoma was lost in follow-up. Although we wanted to examine potential aggressiveness due to tumor size and potential long-term recurrence, our data are too limited to evaluate a possible correlation between the size of the adenoma and the outcome of the disease [5,12,16,18].

In our study, there was no significant association between preoperative ACTH value and disease remission; according to some authors, an increase in this value would reflect the size of the tumor rather than an increase in ACTH [4]. 

Using the cut-off value of 138 nmol/L, it appears that 81.25% of our patients achieved early remission after surgery. Early cortisolemia showed a statistically significant difference between the remission group and the persistence group (*p* < 0.001). Our result is compatible with those in the literature regarding early remission having low value as a predictive factor in long-term follow-up (59–132 months) [3,4,5,7,16,32,34,35].

However, an excellent finding is that our study differs from the literature, in that four patients (12.5%) who presented with non-reduced postoperative cortisol levels showed a subsequent remission (so-called late-remissions—rare but not to be overlooked). It was found that in these patients with early cortisol values >138 nmol/L, there was a gradual remission; one of these patients subsequently relapsed in the follow-up period, underscoring the need for continued vigilance in monitoring cortisol levels post-surgery. Overall, our hypothesis is that while adrenal autonomy is unlikely to be the cause, the exact mechanisms behind the delayed decline in cortisol are still unclear, and may involve factors such as late necrosis of adenoma cells. 

When early hypocortisolemia occurs in the postoperative period, this is considered a positive predictor of complete tumor removal. As a corollary, the lack of early postoperative hypocortisolemia is associated with disease recurrence rates that exceed 67% during the three years following surgery [38]. Although low postoperative cortisol values have a high predictive value for defining the remission of the disease, no value allows the safe exclusion of patients who may develop distant recurrence [16,17,39]. Furthermore, in our study, a patient with early postoperative cortisolemia (<138 nmol/L) had a distant disease recurrence. In the remission group, not all patients had an early postoperative cortisolemia value (<138 nmol/L); in fact, in four cases, the early morning cortisolemia levels dropped later.

One more recommended test that is an accurate identifier of Cushing’s disease is the measurement of urinary-free cortisol [14,19,20,21,22,23,24]. In our patients, 24 h urinary-free cortisol, with measurements beginning immediately after surgery, is significantly associated with disease remission (40.5 [28.5–98] (nmol/24 h), [*p* = 0.019]). As for early 24 h cortisoluria (UFC), this was significantly lower in patients in remission, compared to patients with persistence of disease; however, it does not vary significantly between males and females, or between micro- and macroadenomas. It was not possible to correlate these variables with the risk of long-term recurrence, since only two relapses occurred after initial remission of the disease.

Unlike what has been described by some authors, according to whom the histological confirmation of ACTH-secreting adenoma is associated with a positive prognostic value [4,13,32], in our study, although there was no histological confirmation in 24% of cases, all patients went into remission and were disease-free.

We also observed that in all measurements of cortisolemia and cortisoluria tests, there were no significant changes in cortisol levels during the first five days post-surgery in either the remission or non-remission group. Also, in both groups, there is a positive correlation when values are compared across all five days. Τhe measurements show that in the first five days, the day on which we took the measurements did not play a special role. This means that measurements do not have to be made strictly in the first 48 h, as some authors claim [36].

All patients deemed to be in remission after surgery need lifelong follow-up to identify a possible recurrence, which, in the case of Cushing’s disease, can even occur at 20 years after surgery [4,16,17,32].

### Strengths and Limitations

This is a prospective study in which the evaluation of patients before and after surgery was performed by the same team that conducted the specific study, and the patients with pituitary complications were evaluated as a whole. Indeed, in recurrent diseases, no secondary intervention was followed by our team. One of the potential limitations was the fact that we did not evaluate ACTH concentrations post surgery as a predictor of patient outcome. A limitation in our study was the relatively small number of participants, as ACTH-secreting pituitary adenomas are not common. Its accuracy warrants further validation in a multi-center prospective study with more patients and longer-term follow-up.

## 5. Conclusions

In our cohort of patients with CD, the endoscopic transsphenoidal approach was a safe and effective surgical technique in the treatment of ACTH-secreting pituitary adenoma. Permanent morbidity is rare, and serious complications, although uncommon, can be managed successfully. Early postoperative hypocortisolemia has become an important tool for identifying patients with disease remission after surgery, along with the cortisoluria test. Generally used together with the evaluation of the circadian rhythm, they are valid for use as prognostic factors in identifying patients with persistence and those with recurrence in follow-up. The study reports that 7.6% of patients who initially achieved remission experienced recurrence after 4.5 years. Additionally, 12.5% of patients who did not initially achieve remission (with cortisol levels >138 nmol/L) eventually experienced late remission. 

We found that applying postoperative serum cortisol and cortisoluria within five days can predict subsequent remission with excellent accuracy. Detecting a value of hypocortisolemia ≤138 nmol/L in the first 5 days constitutes an accurate predictor of lasting remission, a distinctive finding in the intermediate term by our team. 

The identification and recognition of recurrence predictors before and after the resection of ACTH-secreting pituitary adenomas represent a clinical challenge. Transsphenoidal surgery can provide rapid symptom relief, and moderate the risk of chronic organ dysfunction resulting from hormonal excess at the same time. Postoperative cortisol levels can possibly be used to show how soon a surgical treatment of the disease will be needed.

## Figures and Tables

**Figure 1 healthcare-12-01900-f001:**
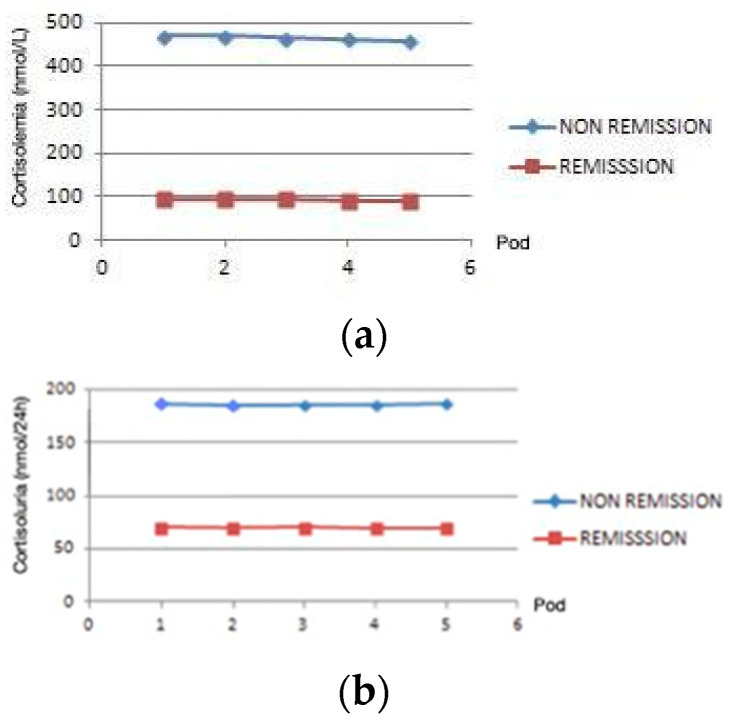
Evaluation of all the examined parameters (blood, urine) from the first to the fifth day from the same patients. (**a**) Postsurgery cortisolemia in nmol/L on the y-axis, and from the first to the fifth day on the x-axis. (**b**) Postsurgery cortisoluria (nmol/24 h) on the y-axis, and from the first to the fifth day on the x-axis. Pod, Postoperative day.

**Table 1 healthcare-12-01900-t001:** Demographic data of the patients.

Qualitative Variable		Number of Patients
**Patients Estimated**		
39 patients	Excluded 6	4 children (3 female and 1 male) and 2 adults (female) with several comorbidities
**Patients Operated**		**Total 33**
Gender	Female	26
	Male	7
Previous surgery in another center	No	26
	Yes	7
MRI identification of the adenoma	No	2
	Microadenoma	23
	Macroadenoma	10
	Diagnosis with IPSS	2
**Age (Median Years)**		**Range**
41.67		**Average** 15–67 years
**Qualitative Variable**		**Mean (SD)**
Pre-surgery UFC/ULN (nmol/24 h)		243.10 (257.23)
Pre-surgery cortisol (nmol/L)		433.04 (170.62)
Pre-surgery ACTH (ng/mL)		73.38 (54)

IPSS, inferior petrosal sinus sampling; SD, standard deviation; UFC, urinary-free cortisol; ULN, upper limit normal level; ACTH, adrenocorticotropic hormone.

**Table 2 healthcare-12-01900-t002:** Biochemical data analyzed by gender.

	Gender		
	Female	Male	
N = 33	(N = 26)	(N = 7)	*p* Value
**Age at Surgery (Years)**			
N	27	7	
Median (IQR)	42.0 (35.0–47.0)	42.0 (30.0–59.0)	0.71
**Months of Follow-Up**			
N	26	7	
Median (IQR)	81.7 (59.5–132.0)	60.0 (60.0–128.0)	0.15
**ACTH Preoperative (ng/mL)**			
N	26	7	
Median (IQR)	50.5 (39.0–69.0)	112.0 (72.0–133.0)	0.023 *
**Morning ** **Cortisol Preoperative (nmol/L)**			
N	26	7	
Median (IQR)	452.0 (392.0–483.0)	552.0 (461.0–611.0)	0.21
**UFC Preoperative (nmol/24 h)**			
N	26	7	
Median (IQR)	141.0 (106.0–238.0)	224.0 (177.0–430.0)	0.24
**Cortisol Postoperative (nmol/L)**			
N	26	7	
Median (IQR)	68.5 (45.0–320.0)	62.0 (34.0–128.0)	0.55
**UFC Postoperative (nmol/24 h)**			
N	26	7	
Median (IQR)	72.0 (32.0–227.0)	35.0 (27.0–57.0)	0.11

IQR, interquartile range (calculated as the difference between Q3 and Q1 of a dataset); UFC, urinary-free cortisol; *, statistical significance.

**Table 3 healthcare-12-01900-t003:** Biochemical data analyzed by remission and non-remission of the patients.

Outcome of Surgery			
	Non-Remission	Remission	
N = 32	(Recurrence)	(N = 26)	*p* Value
	(N = 6)		
**Age at ** **S** **urgery (years)**			
N	6	26	
Median (IQR)	48.5 (42.0–53.0)	41.0 (31.0–47.0)	0.21
**Months of ** **F** **ollow-up**			
N	6	26	
Median (IQR)	75.5 (59.5–130.0)	81.7 (59.0–132.0)	0.64
**ACTH ** **P** **reoperative (ng/mL)**			
N	6	26	
Median (IQR)	50.0 (39.0–100.0)	56.0 (46.0–101.0)	0.59
**Morning ** **C** **ortisol ** **P** **reoperative (nmol/L)**			
N	6	26	
Median (IQR)	486.0 (475.0–541.0)	441.0 (381.0–475.0)	0.13
**UFC ** **P** **reoperative (nmol/24 h)**			
N	6	26	
Median (IQR)	176.0 (130.0–430.0)	152.0 (106.0–242.0)	0.57
**Cortisol ** **P** **ostoperative (nmol/L)**			
N	6	26	
Median (IQR)	453.5 (350.0–521.0)	58.5 (43.0–100.0)	<0.001 **
**UFC ** **P** **ostoperative (nmol/24 h)**			
N	6	26	
Median (IQR)	185.5 (72.0–315.0)	40.5 (28.5–98.0)	0.019 *

IQR, interquartile range (Q1–Q3); UFC, urinary-free cortisol; *, statistical significance; **, statistically highly significant.

**Table 4 healthcare-12-01900-t004:** MRI identification of the adenoma.

N = 33	Microadenoma (N = 23)	Microadenoma (N = 10)	*p* Value
**Age at surgery (years)**			
N	23	10	
Median (IQR)	40.0 (35.0–44.0)	50.0 (30.0–59.0)	0.13
**Months of follow-up**			
N	23	10	
Median (IQR)	79.0 (59.0–132.0)	81.5 (61.0–128.0)	0.28

IQR, interquartile range (Q1–Q3).

**Table 5 healthcare-12-01900-t005:** Examined cortisolemia and cortisoluria, in the remission and non-remission group from the first to the fifth day.

Outcome Measures	Postoperative				
	POD-1	POD-2	POD-3	POD-4	POD-5
**Cortisolemia** **(nmol/L)**					
Remission	95.81(108.8)	95.70(108.81)	95.22(108.21)	91.55(106.98)	89.48(105.99)
Non-Remission	471.16(134.39)	470.66(134.32)	466.83(135.28)	462.83(132.08)	459.53(131.41)
**UFC (nmol/24 h)**					
Remission	70.5(71.66)	70.05(70.76)	70.45(70.86)	69.75(71.22)	69.93(70.67)
Non-Remission	186.66(121.68)	184.83(120.63)	185.33(121.25)	185.33(122.32)	186.83(121.61)

POD, postoperative day; UFC, urinary-free cortisol.

**Table 6 healthcare-12-01900-t006:** Examined parameters (blood, urine) from the first to the fifth day, and possible correlations between POD 1–2, POD 3–4, and POD 1–5.

**Cortisolemia**	**Remission**	**Non-Remission**
Whole population		
POD (1–2): rs = 1, n = 33, *p* = 0.000	POD (1–2): rs = 1, n = 27, *p* = 0.000	POD (1–2): rs = 1, n = 6, *p* = 0.000
POD (3–4): rs = 0.995, n = 33, *p* = 0.000	POD (3–4): rs = 0.993, n = 27, *p* = 0.000	POD (3–4): rs = 1, n = 6, *p* = 0.000
POD (1–5): rs = 0.996, n = 33, *p* = 0.000	POD (1–5): rs = 0.997, n = 27, *p* = 0.000	POD (1–5): rs = 1, n = 6, *p* = 0.000
**Cortisoluria**	**Remission**	**Non-** **remission**
Whole population		
POD (1–2): rs = 0.999, n = 33. *p* = 0.000	POD (1–2): rs = 0.998, n = 27, *p* = 0.000	POD (1–2): rs = 1, n = 6, *p* = 0.000
POD (3–4): rs = 1, n = 33, *p* = 0.000	POD (3–4): rs = 0.999, n = 27, *p* = 0.000	POD (3–4): rs = 1, n = 6, *p* = 0.000
POD (1–5): rs = 0.999, n = 33, *p* = 0.000	POD (1–5): rs = 0.997, n = 27, *p* = 0.000	POD (1–5): rs = 1, n = 6, *p* = 0.000

rs, Spearman’s rank correlation coefficient; n, numbers.

**Table 7 healthcare-12-01900-t007:** Clinical, laboratory and radiologic characteristics of late-remission patients.

ID	Age	Gender	ACTH Pre-Surgery (ng/mL)	UFC Pre-Surgery (nmol/24 h)	Cortisolemia Pre-Surgery (nmol/L)	Previous Surgery	MRI Size	UFC Pod1-Pod2 (nmol/24 h)	Cortisolemia Pod1-Pod2 (nmol/L)	3 Months Cortisolemia (nmol/L)	3 Months UFC (nmol/24 h)	Recurrence at Last Follow-Up
1	35	F	65.30	138	390	No	Micro	94.3	134.39	98.32	35.9	No
2	42	F	56.29	141	368	No	Micro	110.9	160.45	110.23	87.74	Yes
3	45	M	68.20	120	340	No	Micro	90.2	128.29	95.2	85.64	No
4	56	F	61.29	136	385	No	Micro	120.2	135.40	85.67	62.56	No

Micro, microadenoma; Pod, post-operative day; UFC, urinary-free cortisol.

## Data Availability

The data presented in this study are available on request from the corresponding author.

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
