# Peer review of "Prospective Evaluation of Transsphenoidal Pituitary Surgery in Patients with Cushing’s Disease: Delayed Remission and the Role of Postsurgical Cortisol as a Predictive Factor"

_healthcare, 2024, doi:10.3390/healthcare12181900_

Round 1
Reviewer 1 Report
Comments and Suggestions for Authors
The article “Prospective evaluation of transsphenoidal pituitary surgery in 2 patients with Cushing’s syndrome: delayed remission and the 3 role of the postsurgical cortisol as predictive factor”. The topic is interesting, sounds well to readers, and is suitable for the journal.
Well, the manuscript is good, and the author, please incorporates my suggestion to enhance the effectiveness of this manuscript for the esteemed journal and scientific world. All comments are shared below:
Comment 1: Kindly look about the Grammer mistake in full manuscript.
Comment 2: Author writes well but discussion part can be strong
Comment 3: Rewrite the conclusion section.
Comment 4 Rewrite the abstract in scientific term
Comments on the Quality of English Languagemoderate
Author Response
• Comment 1: Kindly look about the Grammer mistake in full manuscript.
We apologize for the mistakes. We have improved the grammar in the new version, correc9ng the mistakes.
• Comment 2: Author writes well but discussion part can be strong
We have modified this sec9on based on the sugges9ons also given by the other reviewers.
• Comment 3: Rewrite the conclusion secBon.
We have modified this sec9on based on the sugges9ons also given by the other reviewers.
• Comment 4: Rewrite the abstract in scienBfic term We have modified the sec9on based on the sugges9on. We are at your disposal for any sugges9ons.

Reviewer 2 Report
Comments and Suggestions for Authors
This study describe the post-operative effectiveness and remission rate in Cushing's syndrome. Overall this study looks important in the context of disease, however many of the results presented in this study looks confusing or have not presented or discussed properly. I have therefore following comments and suggestions for the authors to improve this manuscript.
Major comments:
1. The authors mentioned that a total of 7 patients have underwent previous surgery in another center. Whether these patients are on any drugs to control their cortisol levels were not indicated which might effect the post-operative biochemical measurements taken at this center. Also the information regarding how much long before they have previous surgery would be more helpful.
2. The authors mentioned that four patients who presented with non-reduced postoperative cortisol levels showed a subsequent late remission. Are these 4 patients belong to the same 7 patient group who underwent previous surgery. It is not clear. Also, a bit more explanation on how or why these patients showed subsequent remission would make this study more attractive.
3. The authors mentioned that males have generally high aggressive hormone levels in the cushing syndrome, which is also evident from this study. But it is also clearly visible that both males and females almost have similar post-operative cortisol and UFC levels. Does that mean this surgery manifest a more significant effect on males compared to females. Should be a point in the discussion.
4. In table 3 , the biochemical data as presented by authors didnot suggest any improvement in non-remmision patients category. But the authors claim that 4 of their patients with non-reduced postoperative cortisol levels showed a subsequent remission. A separate table, with biochemical parameters and their previous surgical history will look more convincing.
Some minor comments:
1. Aim of the study should be combined with the introduction- should be the last paragraph in the introduction.
2. The post-operative management section can be trimmed and fined tuned to focus more on methods. For example, how each and every biochemical analysis is performed should be explained more clearly.
Comments on the Quality of English Language
The quality of the English is okay, but should be improved at some places especially when writing legends for figures and tables.
Author Response
See the file "rev 2 .pdf" attached.

Reviewer 3 Report
Comments and Suggestions for Authors
Dear Authors,
I have reviewed your manuscript and found it to be an interesting study with a respectable number of patients, especially considering the rarity of the disease. However, there are several issues that I have identified.
Firstly, the manuscript is sometimes difficult to read, and the information is not completely well organized within the various subsections. Additionally, there are some statistical issues and problems with the organization of the results that might need your attention.
I also recommend a thorough revision of the discussion section, as it contains limited discussion with the work of other authors.
In the attached pdf you will find a detailed list of the issues identified and several doubts.
Thank you very much for the time expended in the writing of this manuscript.
Best regards

Author Response
See the file "rev 3 .pdf" attached.

Round 2
Reviewer 3 Report
Comments and Suggestions for Authors
I would like to express my gratitude to the authors for the time and effort they have invested in improving the manuscript. They have done an excellent job incorporating the suggested changes, and I have very few remaining concerns.
However, I am still having difficulty understanding the information presented in Table 5. The p-values suggest significant differences between P1 and P2 (the first and second day). According to these p-values, the values from different days should be significantly different. Nonetheless, the table and the text indicate that cortisol concentrations within each group remain relatively stable. It’s possible that the p-values are comparing results between recurrence and non-recurrence, but this is not clear. I apologize for insisting on this point, but I am having trouble interpreting the table.
Additionally, the graphs in Figure 1 are missing axis labels. While the text mentions this information, it would be clearer if the labels were included directly in the figure. There also appears to be a formatting issue, as the graph sizes are inconsistent and quite small, which makes the figure look somewhat disorganized. I recommend that the authors address these formatting concerns with the editorial services.
Overall, I believe the manuscript is nearly ready for publication with just a few minor revisions.
Thank you for your time and for allowing me to participate in this review process.
Author Response
Comments to the Author
- Comment 1: I am still having difficulty understanding the information presented in Table 5. The p-values suggest significant differences between P1 and P2 (the first and second day). According to these p-values, the values from different days should be significantly different. Nonetheless, the table and the text indicate that cortisol concentrations within each group remain relatively stable. It’s possible that the p-values are comparing results between recurrence and non-recurrence, but this is not clear. I apologize for insisting on this point, but I am having trouble interpreting the table.
We apologize for the incorrect arrangement of the data. There was probably a misunderstanding. We removed the correlation section with the p values ​​between the days in this table (columns on the right), because it gave a wrong conclusion. Thus, it is clearly seen that the concentrations between the intervening days were relatively stable. Ηowever p- values indicated that there was a positive correlation between days (Positive correlations are indicated between zero and +1. Values of more than zero, i.e. those with a plus value, show a positive correlation. This means that as one value rises the other value rises or as one value falls the other value falls.)
- Comment 2: the graphs in Figure 1 are missing axis labels. While the text mentions this information, it would be clearer if the labels were included directly in the figure. There also appears to be a formatting issue, as the graph sizes are inconsistent and quite small, which makes the figure look somewhat disorganized. I recommend that the authors address these formatting concerns with the editorial services
I think it's a formatting error. The original figures are attached. Thanks for the warning.
